# Double-Stranded RNA Enhances Matrix Metalloproteinase-1 and -13 Expressions through TLR3-Dependent Activation of Transglutaminase 2 in Dermal Fibroblasts

**DOI:** 10.3390/ijms23052709

**Published:** 2022-02-28

**Authors:** Ah-Young Hong, Seok-Jin Lee, Ki Baek Lee, Ji-Woong Shin, Eui Man Jeong, In-Gyu Kim

**Affiliations:** 1Department of Biochemistry and Molecular Biology, Seoul National University College of Medicine, Seoul 03080, Korea; ay630@naver.com (A.-Y.H.); ylsj111@naver.com (S.-J.L.); lkb3431@snu.ac.kr (K.B.L.); earthbear3@gmail.com (J.-W.S.); 2Laboratory for Cellular Response to Oxidative Stress, Cell2in, Inc., Seoul 03127, Korea; 3Department of Pharmacy, College of Pharmacy, Jeju National University, Jeju 63243, Korea; euimanjeong@gmail.com; 4Department of Human-Environment Interface Biology, Seoul National University College of Medicine, Seoul 03080, Korea

**Keywords:** transglutaminase 2, double-stranded RNA (dsRNA), matrix metalloproteinases (MMP), dermal fibroblasts, poly(I:C), ROS

## Abstract

UV-irradiation induces the secretion of double-stranded RNA (dsRNA) derived from damaged noncoding RNAs in keratinocytes, which enhance the expression of matrix metalloproteinases (MMP) in non-irradiated dermal fibroblasts, leading to dysregulation of extracellular matrix homeostasis. However, the signaling pathway responsible for dsRNA-induced MMP expression has not been fully understood. Transglutaminase 2 (TG2) is an enzyme that modifies substrate proteins by incorporating polyamine or crosslinking of proteins, thereby regulating their functions. In this study, we showed that TG2 mediates dsRNA-induced MMP-1 expression through NF-κB activation. Treatment of poly(I:C), a synthetic dsRNA analogue binding to toll-like receptor 3 (TLR3), generates ROS, which in turn activates TG2 in dermal fibroblast. Subsequently, TG2 activity enhances translocation of p65 into the nucleus, where it augments transcription of MMP. We confirmed these results by assessing the level of MMP expression in *Tlr3*^−/−^, TG2-knockdowned and *Tgm2*^−/−^ dermal fibroblasts after poly(I:C)-treatment. Moreover, treatment with quercetin showed dose-dependent suppression of poly(I:C)-induced MMP expression. Furthermore, ex vivo cultured skin from *Tgm2*^−/−^ mice exhibited a significantly reduced level of MMP mRNA compared with those from wild-type mice. Our results indicate that TG2 is a critical regulator in dsRNA-induced MMP expression, providing a new target and molecular basis for antioxidant therapy in preventing collagen degradation.

## 1. Introduction

Skin wrinkling is a prominent feature of aged skin and thought to be caused by an increase of reactive oxygen species (ROS) production due to mitochondrial dysfunction of senescent dermal fibroblast [1]. Oxidative stress induces upregulation of matrix metalloproteinase-1 (MMP-1), an enzyme responsible for degradation of type I collagen, a major component of dermal extracellular matrix (ECM). Moreover, oxidative stress also reduces the synthesis of type I collagen in dermal fibroblasts leading to dysregulation of ECM homeostasis and decreased dermal thickness [2]. In sun-exposed skin, UV irradiation is known to induce ROS generation such as singlet oxygen (^1^O_2_), superoxide anion (O_2_^−^), hydrogen peroxide (H_2_O_2_), and hydroxyl radicals (OH·), and thus aggravates the dysregulation of ECM remodeling through MMP-1 expression, resulting in premature skin aging [3]. However, the signaling pathway involved in the regulation of MMP-1 expression in UV-irradiated or oxidatively stressed dermal fibroblasts has not been fully understood.

Double stranded RNAs (dsRNA) are generated in virus-infected cells during its replication, and play a role in eliciting anti-viral responses by binding toll-like receptor 3 (TLR3), a member of the pattern recognition receptor family, and retinoic acid-inducible gene I (RIG-I)-like receptors (RLRs) [4]. TLR3 and RLRs then activate various signaling pathways, such as NF-ĸB, and MAPK signaling, leading to production of proinflammatory cytokines and induction of interferon-stimulated genes through transcriptional regulation [5]. Interestingly, dsRNA is also produced in stressed, apoptotic or necrotic cells. A previous report showed that UV-irradiation damages noncoding RNAs in keratinocytes, producing and releasing dsRNA, which triggers non-irradiated cells to induce the expression of TLR3-dependent inflammatory cytokines, such as TNF-α and IL-6, thereby eliciting skin inflammatory reactions [6]. Moreover, poly(I:C) induces MMPs’ expression in human dermal fibroblasts [7]. These findings indicate that dsRNA functions as a signaling molecule to induce skin responses in response to genotoxic stresses in addition to viral infection.

Transglutaminase 2 (TG2) is a member of the enzyme family that modifies substrate proteins by catalyzing polyamine incorporating or ε(γ-glutamyl) lysine protein crosslinking reactions [8]. These modifications affect the activity of substrate proteins by altering protein–protein interaction or their solubility [9,10]. TG2 is unique among TG isoenzymes in that intracellular TG2 is activated in response to various stresses, such as UV-irradiation or treatment with DNA-damaging chemotherapeutics, otherwise held in latent inactive enzyme [11,12]. UV-irradiation activates TG2 by one of two mechanisms: (i) by releasing endoplasmic reticulum (ER) calcium [13]; (ii) by activating TGFβ signaling pathway [11] or possibly redox signaling pathway, depending on the level of oxidative stress. Moreover, TG2 is not activated when treated with actinomycin D after oxidative stress, suggesting that TG2 is activated by TGFβ or redox signaling-mediated regulation of gene expression. In the skin, UV-irradiation activates intracellular TG2 in epidermal keratinocytes which enhances NF-ĸB transcriptional activity and production of proinflammatory cytokines, leading to acute inflammation [13]. Recently, we have shown that TG2 mediates UV-induced MMP-1 expression through NF-ĸB activation in dermal fibroblasts [14]. Thus, TG2 is a critical regulator in UV-induced skin inflammation and ECM remodeling.

UV-irradiation elicits skin inflammation and ECM remodeling through activation of various signaling pathways, where both dsRNA and TG2 participate as a mediator, respectively. In this study, we have investigated the possibility that dsRNA regulates TG2 in the UV-induced MMP expression in dermal fibroblasts, and showed that dsRNA activates TG2 through TLR3-dependent ROS production, resulting in enhancement of NF-ĸB signaling and subsequent MMP-1 and -13 expression. Our results identify dsRNA—TLR3—TG2—NF-ĸB as an effector pathway responsible for ECM remodeling in UV-exposed skin, providing new targets to prevent skin wrinkling.

## 2. Results

### 2.1. Poly(I:C) Enhances MMP-1 and -13 Expression through TG2 Activation

To test whether TG2 is involved in the regulation of dsRNA-induced MMP expression, we examined the effect of TG2 knock-down on the levels of MMP expression. In human dermal fibroblasts (HDFs), treatment with poly(I:C), a synthetic dsRNA analog, caused a time-dependent increase in the levels of *MMP1* and *13* mRNA up to 24 h (Figure 1A). In TG2 knock-downed HDFs, levels of *MMP1* and *13* mRNA were significantly reduced in response to poly(I:C) treatment compared with those of control siRNA-treated cells (Figure 1B). Moreover, Western blot analysis showed that TG2 knock-down reduces the level of poly(I:C)-induced MMP-1 protein compared with that of control cells. Notably, poly(I:C) treatment upregulated the protein level of TG2 (Figure 1C). To corroborate these results, we compared collagenase activity of MMPs in both cells using DQ-collagen type 1 as a substrate [15]. Poly(I:C) treatment significantly increased the intensity of DQ-fluorescence in control siRNA-treated cells compared with saline-treated cells, but had no effect in TG2 knock-downed cells (Figure 1D). Furthermore, treatment with cysteamine, which is a non-specific inhibitor, though TG2 is the only isoform expressed in dermal fibroblasts, reduces the level of poly(I:C)-induced *MMP1* and *13* mRNA (Figure 1E). These results indicate that TG2 mediates poly(I:C)-induced MMP expression.

To confirm these results, we compared the ability of wild-type and *Tgm2*^−/−^ mouse dermal fibroblasts (MDFs) to promote Mmp13 expression in response to poly(I:C) treatment. In MDFs prepared from wild-type mice, poly(I:C) treatment resulted in a dose and time-dependent increase of *Mmp13* mRNA peaked at 20 μg/mL and after 12 h, respectively (Figure 2A). Under these experimental conditions, by contrast, *Tgm2*^−/−^ MDFs showed significantly reduced level of and Mmp13 mRNA and protein compared with wild-type MDFs (Figure 2B). These results were verified by measuring collagenase activity. Poly(I:C) treatment induced a significant increase of collagenase activity in wild-type MDFs compared with saline-treated cells, but had no effect on that of *Tgm2*^−/−^ MDFs (Figure 2C). Intracellular TG2 is known to be inactive under physiological culture conditions [11]. We then established the causal relationship between Tg2 and Mmp13 expression by monitoring Tg2 activity after poly(I:C) treatment. MDFs were exposed to poly(I:C) and then incubated with biotinylated pentylamine (BP). After cell lysis, BP-incorporated proteins were estimated by streptavidin-HRP. Wild-type MDFs showed a gradual increase in the levels of TG activity which peaked at 9 h, but *Tgm2*^−/−^ cells exhibited basal levels of enzyme activity (Figure 2D). Furthermore, Tg2 protein level parallel to the TG activity was gradually increased by poly(I:C) treatment, but is not well-correlated with in situ TG activity. Visualization of intracellular BP-incorporated proteins further confirmed poly(I:C)-induced Tg2 activation (Figure 2E). Together, our results indicate that poly(I:C) upregulates and activates TG2, thereby promoting MMP expression.

### 2.2. ROS Generated by Binding of Poly(I:C) with TLR3 Activates TG2

dsRNA is recognized by TLR3 that transduces signals to IRF3 and NF-κB by recruiting the TIR domain-containing adaptor molecule 1 (TICAM1), leading to induction of type 1 interferon and inflammatory cytokines [16]. To test whether Tlr3 is implicated in the poly(I:C)-induced TG2 activation, we treated wild-type and *Tlr3*^−/−^ MDFs with poly(I:C) and measured intracellular TG activity. Poly(I:C) treatment caused a significant increase of TG activity of wild-type cells but not that of *Tlr3*^−/−^ cells compared with saline-treated cells (Figure 3A), indicating that Tlr3 is required for poly(I:C)-induced TG2 activation. In WT and Tlr3^−/−^ MDFs, poly(I:C) treatment had no effect on the level of Tg2 mRNA and protein compared with saline-treated cells (Figure 3B,C). By contrast, poly(I:C) treatment increased the level of *Tlr3* mRNA (Figure 3D), suggesting that the expression of Tg2 and Tlr3 may be regulated by negative and positive feedback mechanisms, respectively, in poly(I:C)-treated MDFs.

We next investigated the mechanism through which binding of poly(I:C) with TLR3 may affect TG2 activity. It has been reported that activation of TLR3 induces ROS generation by recruiting NOX2 in macrophage [17]. To test whether poly(I:C) generates ROS through TLR3 binding, we compared ROS levels after poly(I:C) treatment in wild-type and *Tlr3*^−/−^ MDFs by measuring oxidation of redox-sensitive fluorescent dye H_2_DCFDA. Flow cytometric analysis showed that control *Tlr3*^−/−^ MDFs exhibit significant low ROS levels compared with control wild-type MDFs. Poly(I:C) treatment had increased the ROS level gradually in both MDFs, but the ROS level in *Tlr3*^−/−^ MDFs still remained lower than that of wild-type MDFs (Figure 4A). These results suggest that poly(I:C) treatment generates ROS by binding to TLR3 and by unknown mechanism. Under the same experimental conditions, in contrast, there was no significant difference in fluorescence intensity between wild-type and *Tgm2*^−/−^ MDFs (Figure 4B), indicating that poly(I:C) generates ROS by binding with TLR3.

To test whether poly(I:C)-generated ROS activates TG2, we firstly assessed the effect of tert-butyl hydroperoxide (tBHP) treatment on TG activity. Treatment with tBHP resulted in a significant increase of intracellular TG activity in wild-type MDFs, but not in *Tgm2*^−/−^ cells (Figure 4C). When cells were treated with poly(I:C), a similar increase of TG activity was observed in wild-type MDFs. In contrast, co-treatment with quercetin decreased TG activity to the baseline levels. Both poly(I:C) or quercetin treatment showed no effect on TG activity in *Tgm2*^−/−^ cells (Figure 4E). Under these conditions, we also examined the effect tBHP and poly(I:C) on MMP-13 expression. Treatment with tBHP induced a dose-dependent increase of *Mmp13* mRNA in wild-type MDFs, but it had no effect on the levels of *Mmp13* expression in *Tgm2*^−/−^ cells (Figure 4D). Similarly, poly(I:C) treatment increased the level of *Mmp13* mRNA in wild-type MDFs, whereas co-treatment with quercetin decreased the level of poly(I:C)-induced *Mmp13* expression in a quercetin dose-dependent manner. The effect of quercetin co-treatment was not observed in *Tgm2*^−/−^ cells (Figure 4F). Thus, our results indicate that poly(I:C)-induced ROS upregulate Mmp13 expression by activating TG2 in MDFs.

### 2.3. TG2 Mediates Poly(I:C)-Induced NF-kB Activation

Expression of MMPs is known to be regulated by NF-κB and AP-1 transcription factor [18], and dsRNA induces activation of NF-κB, JNK and p38 through TLR3 [19]. To determine which signaling pathway(s) is/are activated by TG2 for MMP13 expression, we compared the ability of wild-type and *Tgm2*^−/−^ MDFs to enhance AP-1 and NF-κB dependent transcription in response to poly(I:C) treatment. In MDFs transfected with AP-1 luciferase reporter, there was no difference in AP-1 reporter activity between wild type and *Tgm2*^−/−^ cells, although an increased activity by poly(I:C) treatment was observed in both cells (Figure 5A). In MDFs transfected with 3 κB luciferase reporter, in sharp contrast, poly(I:C) treatment caused an approximately eightfold increase of reporter activity in wild-type cells but not in *Tgm2*^−/−^ MDFs, compared with saline-treated cells (Figure 5B). To confirm these results, we compared protein levels of c-jun and p65 in wild-type and *Tgm2*^−/−^ MDFs after poly(I:C)-treatment. Western blot analysis showed an increased level of c-jun protein in both cells, but there was no difference between wild-type and *Tgm2*^−/−^ cells. By contrast, poly(I:C) treatment enhanced the phosphorylation of p65 at Ser 536 in wild-type MDFs, but not in *Tgm2*^−/−^ cells (Figure 5C). When cells were separated into nuclear and cytoplasmic fractions, the level of p65 in the nucleus was significantly increased in wild-type MDFs, but not in *Tgm2*^−/−^ cells, compared with saline-treated MDFs (Figure 5D), indicating that TG2 is responsible for poly(I:C)-induced NF-kB activation leading to MMP13 expression.

We next corroborated the role of TG2 in vivo, and to this end we used ex vivo skin culture model [14,20]. Skin explants were prepared from neonatal mice and cultured with various doses of poly(I:C) for 24 h. Western blot analysis showed a dose-dependent increase of Mmp13 protein (Figure 6A). In skin explants from wild-type mice, poly(I:C) significantly increased the level of Mmp13, whereas in skin explants from *Tgm2*^−/−^ mice poly(I:C) had no effect compared with saline-treated skin explants (Figure 6B). Under these conditions, we assessed the level of intracellular TG activity in the skin explants by measuring BP incorporated into cellular protein for 1 h prior to harvest. In skin explants from wild-type mice, poly(I:C) treatment caused an approximately three-fold increase in TG activity, while poly(I:C) had minimal effect in skin explants from *Tgm2*^−/−^ mice (Figure 6C). These results further support a critical role of TG2 in the regulation of poly(I:C)-induced MMP expression.

## 3. Discussion

UV irradiation increases MMPs’ expression in dermal fibroblasts that leads to fragmentation and loss of collagen fibrils, the most abundant ECM protein in skin dermis, resulting in thin and fragile skin [21,22,23,24]. Previous reports showed that dsRNA is generated in the skin following UV-irradiation, and is responsible for enhancing MMPs’ expression, thereby inducing alterations of dermal ECM [2,7]. Although the NF-κB and AP-1 signaling pathways are involved in MMPs’ expression, how dsRNA activates these pathways is not fully understood. Previously, we showed that TG2 is activated under various oxidative stress conditions, including UV irradiation, ROS-generating chemotherapeutics and H_2_O_2_ treatment [11,12,13,25,26,27]. Moreover, poly(I:C) treatment generates ROS through TLR3 binding in macrophages [17]. In this study, therefore, we tested whether poly(I:C) treatment could activate TG2. Our results showed that poly(I:C) generates ROS by binding to TLR3 and by an unknown mechanism, which activates TG2 in dermal fibroblast. Subsequently, TG2 activity enhances translocation of p65 into the nucleus, where it augments transcription of MMPs. These results indicate that TG2 serves as an effector enzyme that links dsRNA and NF-κB activation in dermal fibroblasts, resulting in ECM dysregulation by promoting MMPs’ expression.

TLR3, a member of the pattern recognition receptor family, specifically recognizes dsRNA produced in virus-infected cells. In immune cells, dsRNA binding to TLR3 activates IRF3 and NF-κB through MyD88-independent pathway, inducing the expression of type I interferon and proinflammatory cytokines that exert a potent antiviral immunity [16,28]. Mechanistically, TLR3 recruits TICAM1 which undergoes oligomerization and in turn recruits TBK1/IKKε and RIP1/TAK1, leading to activation of IRF3 and NF-κB, respectively [29]. However, the TLR3 signaling pathway in dermal fibroblasts has not been elucidated. The results described here showed that poly(I:C) induces TLR3-dependent ROS production which causes TG2 activation, leading to MMPs’ expression through NF-κB activation. In support of these results, previous reports showed that wild-type mice receiving TG2-deficient bone marrow cells exhibit a similar inflammatory response to bleomycin, a ROS producing chemotherapeutics [30], or imiquimod, a ligand of TLR7, compared with wild-type mice [26,27]. Furthermore, TG2 is dispensable for differentiation of T-cell and macrophage [27]. Thus, our findings provide a new mechanism for dsRNA-induced NF-κB activation in non-immune cells.

UV irradiation induces the formation of DNA lesions, such as pyrimidine dimer, that activates signaling pathways participated in DNA repair and cell cycle control, leading to cell survival or apoptosis [31,32]. Moreover, keratinocytes secrete a variety of bioactive molecules that are upregulated by p53-dependent transcriptional control, and mediate autocrine and paracrine DNA damage responses. For instance, α-MSH, a cleavage product of p53-induced pro-opiomelanocortin, is secreted from UV-exposed keratinocytes, and stimulates melanin synthesis in melanocytes, contributing to protection from UV-induced DNA damage [33]. Similarly, dsRNA derived from double-stranded RNA domains of damaged cellular RNA is secreted from UV-exposed keratinocytes, and in turn induces paracrine production of various cytokines, including TNF-α and IL-6, thereby eliciting acute inflammation of the skin and systemic immune suppression [6]. Our results have shown that dsRNA is one of the molecules responsible for UV-induced upregulation of MMP expression through activation of NF-κB in dermal fibroblasts. Combined with the similar paracrine NF-κB activation of dsRNA in keratinocytes, our findings suggest that dsRNA released from keratinocytes functions as a signaling molecule for DNA damaging responses in dermal skin.

TGFβ play a key role in maintaining dermal ECM [34,35]. Binding to its cell surface receptor TβRII-TβRI transmits signals by phosphorylating Smad2/3 which recruits Smad4. The Smad complex are then translocated into the nucleus, where it enhances the transcription of ECM proteins [36]. Previously, we have reported that TGFβ signaling activates TG2 in a Smad-dependent manner [37]. In the skin, however, UV-irradiation suppresses TGFβ signaling by reducing TβRII expression [38]. Thus, our results indicate that dsRNA-induced TG2 activation is not dependent on TGFβ signaling, but on TLR3 signaling. Moreover, our results provide conflicting evidence on the role of TG2 in the regulation of ECM homeostasis. Whereas TGFβ-dependent TG2 activation resulted in lung fibrosis in bleomycin-treated mice [27], the results described here showed that TLR3-dependent TG2 activation causes collagen degradation in dsRNA-treated skin. In addition, TG2 is dispensable in the development of liver cirrhosis in carbon tetrachloride or thioacetamide-treated mice [39]. By sharp contrast, TG2 contributed to renal fibrogenesis in subtotal nephrectomy or a urethral obstruction model through TGFβ activation [40,41]. These findings suggest that the effect of TG2 activation on ECM is likely to depend on activating agent or fibroblast cell type, although the possibility that other factors, such as inflammatory cytokines, may be involved in the TG2-mediated regulation of both collagen and MMP expression in fibroblasts cannot be excluded. Moreover, externalized TG2-mediated crosslinking of collagen fibrils may limit MMP-1 or -13 activities, resulting in fragmented and coarsely distributed collagen fibrils, as observed in skin chronically exposed to UV radiation.

In the present study, we have investigated a mechanism explaining how dsRNA secreted from keratinocytes in response to UV-irradiation enhances MMP expression in dermal fibroblasts, providing a new target for preventing UV-induced dermal ECM dysregulation. In this respect, it may be worth testing the effect of agents that exhibit inhibitory activity for intracellular TG2 on photoaging of skin.

## 4. Materials and Methods

### 4.1. Cell Culture and Poly(I:C) Treatment

Human dermal fibroblasts (HDFs) were purchased from Gibco. Primary mouse dermal fibroblasts (MDFs) were prepared from neonatal mice skin as previously described [14]. Cells were cultured in Dulbecco’s Modified Essential Medium (DMEM, Welgene, Gyeongsan-si, Korea) supplemented with 10% fetal bovine serum (FBS) (Hyclone), 100 units/mL penicillin and 100 µg/mL streptomycin (PS) (Thermo Fisher Scientific, Waltham, MA, USA). For poly(I:C) (invivogen, San Diego, CA, USA, tlrl-pic-5) treatment, cells were first serum-starved starved for 24 h in DMEM containing 0.1% FBS. Then, cells were treated with saline (for vehicle) or indicated concentrations of poly(I:C) for 12 or 24 h. MDFs or HDFs were used under passage 3 or 8, respectively.

### 4.2. Isolation of Mouse Dermal Fibroblasts and Ex Vivo Mouse Skin Culture

Mouse dermal fibroblasts (MDFs) were prepared from neonatal mice skin as previously described [14]. For ex vivo skin culture, the skin removed from WT or *Tgm2*^−/−^ neonate mouse was incubated in DMEM containing 0.1% FBS and 1% PS for 24 h. Then, the skin was treated with poly(I:C) (50 µg/mL) for next 24 h. All animal experiments were approved by the institutional animal care and use committee (IACUC) of Seoul National University (SNU-160613-4).

### 4.3. siRNA Transfection

For siRNA transfection, GFP (used as control siRNA) and TG2 siRNAs (Santa Cruz Biotechnology, Dallas, TX, USA, sc-45924 and sc-37514) were transfected using RNAiMAX (Invitrogen, Thermo Fisher Scientific, Waltham, MA, USA) according to the manufacturer’s instructions. All experimental procedures were carried out 24 h after transfection.

### 4.4. Total RNA Extraction and qRT-PCR

Total RNA was extracted from cells and mouse skin using an RNA-Spin Total RNA extraction kit (iNtRON biotechnology, Seongnam-si, Korea). Extracted RNA was then converted to cDNA using oligo-dT primers and Superscript II reverse transcriptase (Invitrogen, Thermofisher, Waltham, MA, USA). qRT-PCR was performed with a CFX96 real-time system (Biorad, Hercules, CA, USA) using SYBR qPCR master mix (Kapa Biosystems, Wilmington, MA, USA). The sequences of forward and reverse primers are designed as the following; mouse *36b4* forward, 5′-GAGGCCACACTGAACAT-3′, reverse, 5′-ATGCTGCCGTTGTCAAACAC-3′; *Mmp13* forward, 5′-AGGAAGACCTTGTGTTTGCAGAGC-3′, reverse, 5′-TTCAGGATTCCCGCAAGAGTCG-3′; human *36B4* forward, 5′-GCCAATAGACAGGAGCGCTATC-3′, reverse, 5′-AAAGACGATGTCACTTCCACGAG-3′; *MMP1* forward, 5′-GGTGTCTCACAGCTTCCCAG-3′, reverse, 5′-CCGCTTTTCAACTTCCCTCC-3′. Relative mRNA expression was calculated by 2^−^^ΔΔCt^ method. Gene expression was normalized to mouse *36b4* or human *36B4*.

### 4.5. Western Blot Analysis

Cell lysis, protein sampling and western blot analysis were performed as described previously [42]. Mouse skin samples were homogenized in RIPA buffer (50 mM Tris-Cl, pH 7.4, 150 mM NaCl, 1% NP-40, 1% Sodium deoxycholate and 0.1% SDS) by Ultra-Turrax Dispenser (IKA, Staufen, Germany). The following antibodies were used in this study: anti-MMP13 (Neomarkers, Santa Cruz Biotechnology, Dallas, TX, USA, MS-827-P1) anti-MMP1 (Santa Cruz Biotechnology, Dallas, TX, USA, sc-8834) anti-b actin, anti-p65 (Cell signaling Technology, Danvers, MA, USA, #8242), anti-COX1 (Santa Cruz Biotechnology, Dallas, TX, USA, sc-1752), anti-Lamin B (Santa Cruz Biotechnology, Dallas, TX, USA, sc-6216). Monoclonal antibody against TG2 was prepared as described previously [10]. Immunoblots were quantitated by Image J software (http://rsb.info.nih.gov/ij/, accessed on 1 March 2016).

### 4.6. In Situ Transglutaminase Assay

In situ transglutaminase (TG) assay was measured as described previously [43]. Following 12 h poly(I:C) treatment, cells were incubated with 1mM EZ-link Pentylamine-Biotin (BP) (Thermo Fisher Scientific, Waltham, MA, USA) as a TG substrate in serum-free DMEM for 1 h. These BP-incorporated cells were reacted with Alexa green-conjugated streptavidin (Thermo Fisher Scientific, Waltham, MA, USA) and visualized using FV-1000 confocal laser-scanning microscope (Olympus, Tokyo, Japan).

### 4.7. Cell In Situ Zymography

To measure the MMP activity in cells, Cell in situ zymography was performed and quantified as previously described [14,44]. Briefly, cells were incubated on the sterile cover slip and then treated with 20 µg/mL poly(I:C) for 12 h. The cells were fixed with methanol for 15 min at −20 °C and then, incubated with 10 µg/mL highly quenched FITC-labeled DQ™-collagen type-I (Molecular Probes, Thermo Fisher Scientific, Waltham, MA, USA) at 37 °C for 1h. After incubation and wash six times with 1X PBS, the coverslips were mounted and observed using FV-1000 confocal laser-scanning microscope (Olympus, Tokyo, Japan). Corrected Total Cell Fluorescence (CTCF) were calculated by Image J software (http://rsb.info.nih.gov/ij/, accessed on 1 March 2016).
CTCF = Integrated density of selected cell − (Area of selected cell × Mean fluorescence of background).

### 4.8. Measurement of Intracellular ROS

Intracellular ROS levels were measured by the fluorescence intensity of the 2′7′-dichlorofluroescein diacetate (H_2_DCF-DA) (Thermo Fisher Scientific, Waltham, MA, USA). H_2_DCF-DA was reconstituted in DMSO to make a concentrated stock solution. After 15 min of poly(I:C) treatment, cells pre-treated with or without 20 µg/mL poly(I:C) were incubated in DMEM containing 0.1% FBS for the indicated time. Cells were then treated with 10 µM H_2_DCF-DA and incubated at 37 °C for 15 min prior to harvest. After poly(I:C) treatment for less than 15 min, cells were pre-treated with 10 µM H_2_DCF-DA and then treated with 10 µM H_2_DCF-DA and 20 µg/mL poly(I:C) (or saline) for the indicated time. Cells were washed twice using cold PBS and collected in 2% FBS-PBS. To measure fluorescence intensity, flow cytometry was performed on BD FACS Calibur (BD Biosciences, Franklin Lakes, NJ, USA) and all data were analyzed using Flowjo 7.6 software (BD Biosciences, Franklin Lakes, NJ, USA).

### 4.9. Luciferase Reporter Assay

For AP-1-luciferase and 3κB-luciferase assay, cells were transfected with 3xAP1pGL3 vector or 3kB-luciferase construct using Lipofectamine 3000 (Invitrogen, Thermofisher, Waltham, MA, USA). 3xAP1pGL3 (3xAP-1 in pGL3-basic) was a gift from Alexander Dent (Addgene plasmid # 40342; http://n2t.net/addgene:40342 (accessed on 23 January 2022); RRID:Addgene_40342) [45]. pRL-TK, encoding *Renilla* luciferase, was co-transfected to normalize luciferase activity. After 6 h transfection, cells were incubated in DMEM supplemented with 0.1% FBS and poly(I:C) (20 µg/mL) for 9 h. Luciferase activity was measured using the Dual-Luciferase Reporter Assay System (Promega, Madison, WI, USA) following the manufacturer’s instructions.

### 4.10. Statistical Analysis

GraphPad Prism 5.0 statistical software (GraphPad Software, San Diego, CA, USA) was used for statistical evaluations. One- or Two-way ANOVA was conducted for evaluation. The statistical details can be found in the figure legends. Significance levels were as follows: ***, *p* < 0.05; ****, *p* < 0.01; *****, *p* < 0.001; and n.s., not significant.

## Figures and Tables

**Figure 1 ijms-23-02709-f001:**
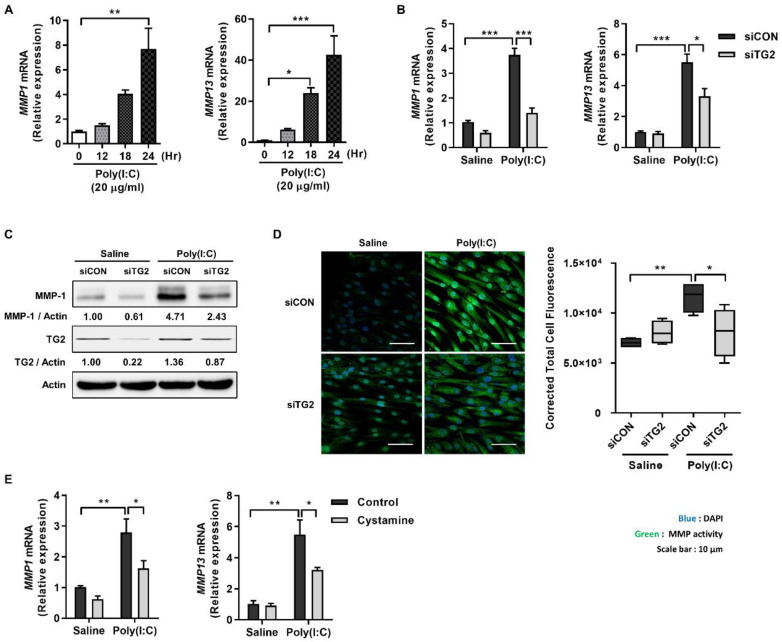
Poly(I:C) enhances MMP-1 and -13 expression through TG2 activation in HDFs. (**A**) HDFs were treated with 20 µg/mL of poly(I:C), and mRNA levels of *MMP1* and *MMP13* were monitored for 24 h using qRT-PCR. (*n* = 3) (**B**,**C**) TG2 expression of HDFs was knock-downed with siRNA. Cells were then treated with 20 µg/mL of poly(I:C) for 24 h. Levels of *MMP1* and *MMP13* mRNA (**B**) and MMP-1 protein (**C**) were measured using qRT-PCR and Western blotting, respectively. (*n* = 3) (**D**) TG2-knockdowned HDFs were treated with 20 µg/mL poly(I:C) for 24 h and then were incubated with 10 µg/mL DQ™-collagen I for 1 h. Intracellular MMPs activity was visualized using confocal microscope (left, scale bar: 10 µm) and quantitated by measuring corrected total cell fluorescence using Image J (right, *n* = 11). (**E**) HDFs were pretreated with cysteamine (100 µM for 1 h) before poly(I:C) treatment (20 µg/mL for 2 h). Levels of *MMP1* and *MMP13* mRNA were measured using qRT-PCR. (*n* = 3) All data are represented as mean ± SEM. Statistical significance was tested by one-way ANOVA by Dunnett’s post-test (**A**) or two-way ANOVA by Turkey’s post-test (**B**,**D**,**E**). ***, *p* < 0.05; ****, *p* < 0.01; *****, *p* < 0.001.

**Figure 2 ijms-23-02709-f002:**
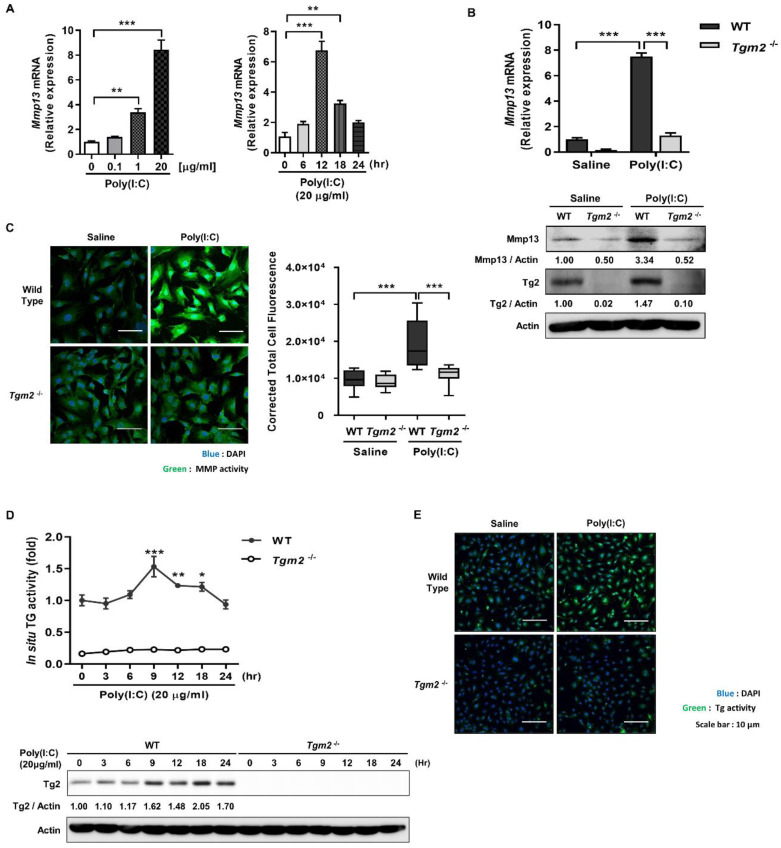
*Tgm2*-deficient MDFs exhibit reduced Mmp-13 expression in response to poly(I:C) treatment. (**A**) MDFs were treated with various concentrations of poly(I:C), and mRNA levels of *Mmp13* were measured after 12 h using qRT-PCR (left, *n* = 3). *Mmp13* mRNA levels of MDFs treated with 20 µg/mL of poly(I:C) were monitored for 24 h (right, *n* = 3). (**B**) *Mmp13* mRNA level (upper panel) and Mmp13 protein (lower panel) of wild-type and *Tgm2*^−/−^ MDFs were measured using qRT-PCR and western blotting, respectively, after treatment with 20 µg/mL of poly(I:C) for 12 h. (*n* = 3) (**C**) Intracellular MMPs activity of wild-type and *Tgm2*^−/−^ MDFs was visualized after treatment with 20 µg/mL of poly(I:C) for 12 h using DQ™-collagen I (left) and quantitated by measuring corrected total cell fluorescence using Image J (right, *n* = 11). (**D**) Wild-type and *Tgm2*^−/−^ MDFs were treated with poly(I:C) (20 µg/mL for 12 h). Intracellular TG activity (upper panel) and Tg2 protein level (lower panel) were monitored for 24 h using biotinylated pentylamine (BP) incorporation assay (*n* = 3) and western blotting. Intracellular TG activity was visualized by detection of BP incorporated proteins using Streptavidin-FITC (**E**). All data are represented as mean ± SEM. Statistical significance was tested by one-way ANOVA by Dunnett’s post-test (**A**) or two-way ANOVA by Turkey’s post-test (**B**–**D**). ***, *p* < 0.05; ****, *p* < 0.01; *****, *p* < 0.001.

**Figure 3 ijms-23-02709-f003:**
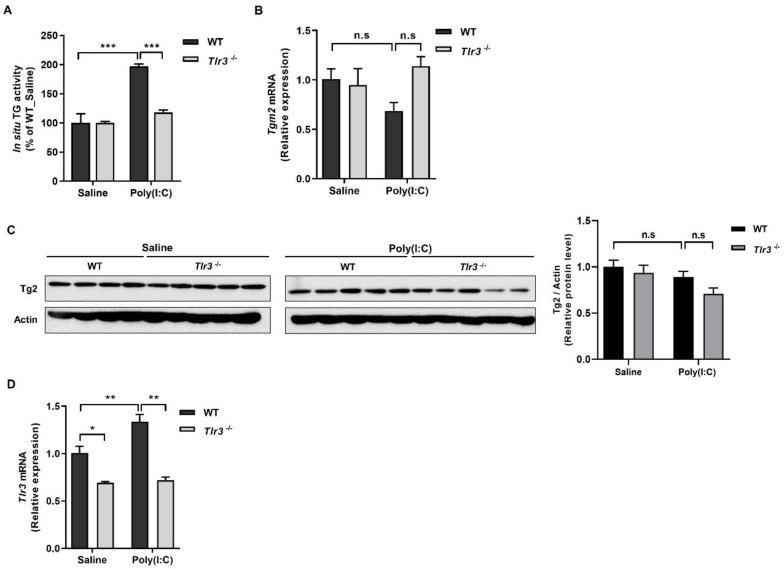
Tlr3 is required for poly(I:C)-induced Tg2 activation in MDFs. (**A**–**D**) Wild-type and *Tlr3*^−/−^ MDFs were treated with 20 µg/mL of poly(I:C) for 9 h. Intracellular TG activity was measured using biotinylated pentylamine (BP) incorporation assay ((**A**), *n* = 3). mRNA level of *Tgm2* and protein level of Tg2 were measured using qRT-PCR ((**B**), *n* = 3) and western blotting (**C**), respectively. The ratio of Tg2 to Actin was estimated by densitometric analysis of Western blot (**C**, right). mRNA level of *Tlr3* were measured using qRT-PCR (**D**). All data are represented as mean ± SEM. Statistical significance was tested by two-way ANOVA by Turkey’s post-test. ***, *p* < 0.05; ****, *p* < 0.01; *****, *p* < 0.001; n.s., not significant.

**Figure 4 ijms-23-02709-f004:**
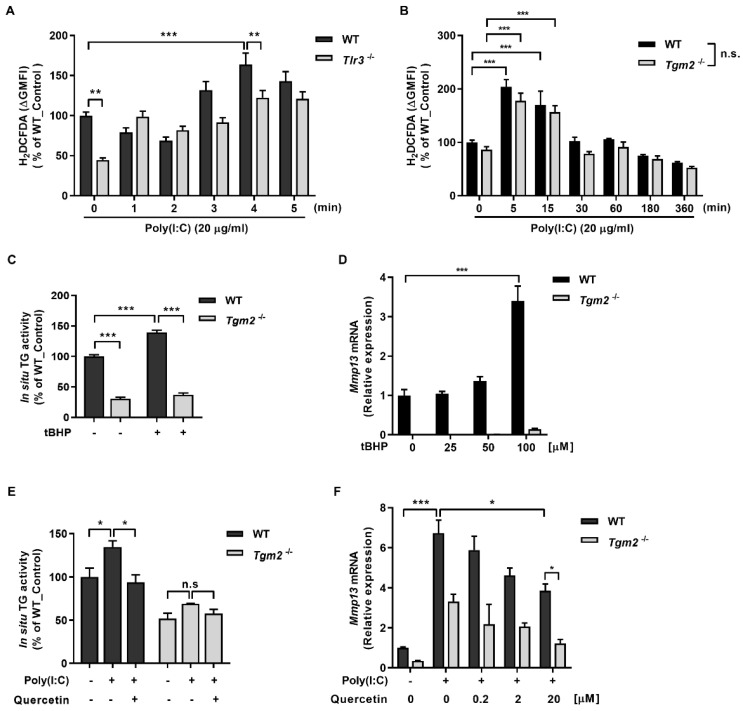
Poly(I:C) binding to TLR3 generates ROS that activates TG2. (**A**,**B**) Wild-type and *Tlr3*^−/−^ (**A**) or *Tgm2*^−/−^ (**B**) MDFs were treated with 20 µg/mL of poly(I:C) and harvested at indicated time. Levels of ROS in the cells were quantitatively measured by flow cytometry using DCFDA. (**C**,**D**) Wild-type and *Tgm2*^−/−^ MDFs were treated with 100 µM (**C**) or various dose (**D**) of tert-butyl hydroperoxide (tBHP) for 12 h. The levels of intracellular TG activity (**C**, *n* = 3) and *Mmp13* mRNA ((**D**), *n* = 3) were evaluated by BP incorporation assay and by qRT-PCR, respectively. (**E**,**F**) Wild-type and *Tgm2*^−/−^ MDFs were treated with 20 µg/mL of poly(I:C) for 12 h. Effect of co-treatment with 20 µM (**E**) or various dose (**F**) of quercetin for 9 h on the levels of intracellular Tg activity ((**E**), *n* = 3) and *Mmp13* mRNA ((**F**), *n* = 3) was evaluated by BP incorporation assay and by qRT-PCR, respectively. All data are represented as mean ± SEM. Statistical significance was tested by two-way ANOVA by Turkey’s post-test. ***, *p* < 0.05; ****, *p* < 0.01; *****, *p* < 0.001; n.s., not significant.

**Figure 5 ijms-23-02709-f005:**
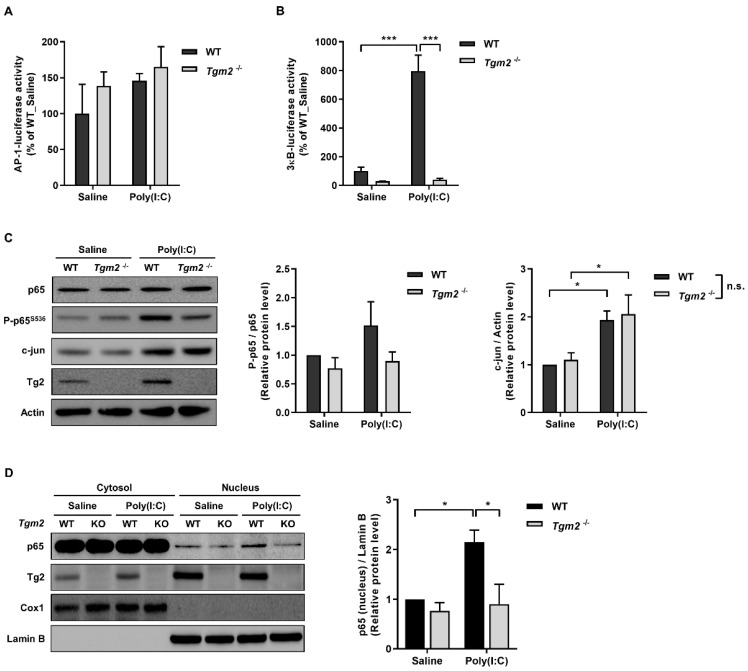
TG2 mediates poly(I:C)-induced NF-κB activation. (**A**,**B**) Wild-type and *Tgm2*^−/−^ MDFs were transfected with expression vector encoding AP-1 (**A**), or 3κB-luciferase reporter (**B**). Luciferase activity was measured after treatment of cells with 20 µg/mL of poly(I:C) for 9 h. (*n* = 3) (**C**) Wild-type and *Tgm2*^−/−^ MDFs were treated with 20 µg/mL of poly(I:C) for 9 h. Total cell extracts were immunoblotted with anti-p65, p-65^S536^ and c-jun antibodies (left). The ratios of p-p65^S536^ to p65 and c-jun to Actin were estimated by densitometric analysis of Western blot (right, *n* = 7). (**D**) Lysates prepared from wild-type and *Tgm2*^−/−^ MDFs treated with 20 µg/mL of poly(I:C) for 6 h were separated into cytoplasmic and nuclear fractions. The nuclear translocation of p65 was assessed by immunoblotted with anti-p65 antibody (left, a representative image). The ratio of p65 to Lamin B was estimated by densitometric analysis of Western blot (right, *n* = 3). Cox-1 and Lamin B were used as an internal control for cytoplasmic and nuclear fraction, respectively. All data are represented as mean ± SEM. Statistical significance was tested by two-way ANOVA by Turkey’s post-test. ***, *p* < 0.05; *****, *p* < 0.001; n.s., not significant.

**Figure 6 ijms-23-02709-f006:**
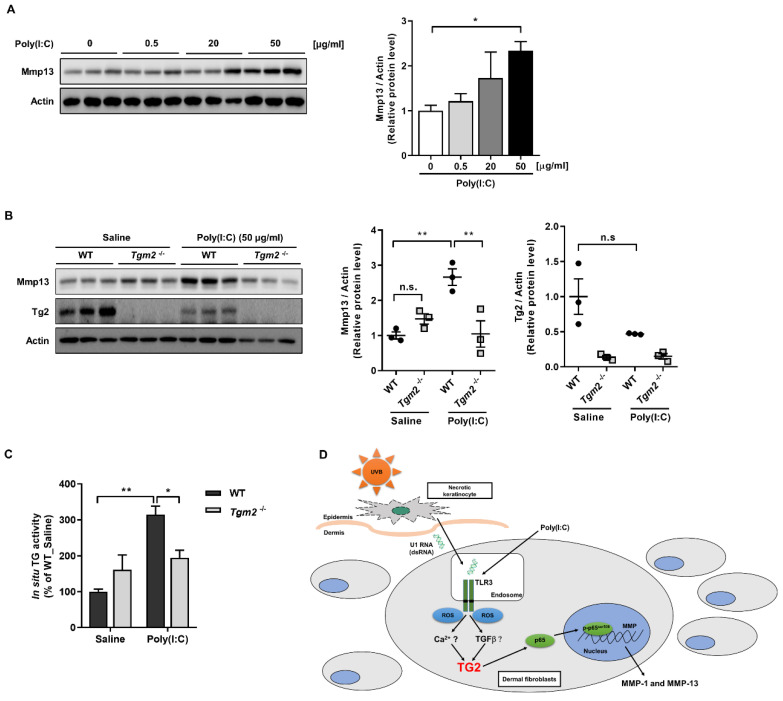
Ex vivo-cultured *Tgm2*^−/−^ mouse skin shows reduced Mmp13 expression in response to poly(I:C) treatment. (**A**,**B**) Skin explants were prepared from wild-type and *Tgm2*^−/−^ neonatal mice treated with various doses (**A**) or 50 µg/mL (**B**) of poly(I:C) for 24 h. Total skin extracts were immunoblotted with anti-Mmp13 antibody (left, *n* = 3). The ratio of Mmp13 or Tg2 to Actin was estimated by densitometric analysis of Western blot (right). (**C**) Effect of poly(I:C) treatment (50 µg/mL for 9 h) on in situ TG activity in wild-type and *Tgm2*^−/−^ mouse skin explants. Intracellular TG activity was assessed by BP incorporation assay. (*n* = 3) (**D**) A schematic representation of poly(I:C)-induced MMPs expression through TG2 activation. All data are represented as mean ± SEM. Statistical significance was tested by one-way ANOVA by Dunnett’s post-test (**A**) or two-way ANOVA by Turkey’s post-test (**B**,**C**). ***, *p* < 0.05; ****, *p* < 0.01; n.s., not significant.

## Data Availability

The data presented in this study are available on request from the corresponding author.

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
