# Peer review of "Double-Stranded RNA Enhances Matrix Metalloproteinase-1 and -13 Expressions through TLR3-Dependent Activation of Transglutaminase 2 in Dermal Fibroblasts"

_ijms, 2022, doi:10.3390/ijms23052709_

Round 1

Reviewer 1 Report

The manuscript of Ah-Young Hong and co-workers describes by the use of TG2-/- and TRL3-/- cells, respectively,  the checkpoint role of transglutaminase 2 (TG2) protein cross-linking activity in the mediation of dsRNA effect in inducing NF-κB, MMP1 and MMp13 activation in dermal fibroblasts which can contribute to ECM damage and ageing in the skin. The presentation of experiments and results is clear and comprehensive, also confirming the main effects in vivo on skin explant models. The results raise the possibility that TG2 activity blocking may have therapeutic advantage to attenuate UV damage in skin.

Comments, questions

  1. The authors suggest that increased MMP1 and MMP13 activation by TG2 primarily leads to ECM degradation and not to fibrosis. However, TG2 can also affect/induce fibrosis in other pathways or by direct enzymatic action on ECM proteins and the relative effects of these consequences is still unkown. Have you observed any signs of fibrosis in dermal explant experiments? Has TG2 externalised in the dermal fibroblast culture? Please discuss in the discussion section.
  2. Reflecting also reference 13 where role of TG2 activation by UV was also shown in keratinocytes, which of the mechanisms (epidermal versus dermal in connective tissue fibroblasts) consider the authors more important for skin damage and how other transglutaminases also present in the skin epidermal layer can be involved as well, since in dermal explant experiments with TG2-/- cells still significant transglutaminase activity was present ?
  3. Please describe more in details the specification (clone name) and binding characteristics of the non-commercial monoclonal anti-transglutaminase-2 antibodies used in the experiments. Which domain of TG2 they recognize? Was cross-reaction with other transglutaminases excluded? Does the antibody recognize both intracellular and extracellular TG2 and in which staining pattern was the TG2 protein expressed in the human dermal fibroblasts under basal conditions and after PolyI:C treatment ? Were any parallel experiments checked with other known commercial anti-TG2 antibodies like e.g. CUB7402 ?
  4. Please specify in the methods that you used FITC-labeled DQ™-collagen type-I for the experiments. Both  DQ™-collagen type-I  and type-IV are commercially available.

Author Response

Feb 20, 2020

Simone Beninati

Guest Editor

Dear Dr. Simone Beninati:

We are pleased to submit to you a revised manuscript entitled “Double-stranded RNA enhances matrix metalloproteinase-1 and -13 expressions through TLR3-dependent activation of transglutaminase 2 in dermal fibroblasts” [Manuscript ID: ijms-1588932]. The manuscript has been changed along the line suggested by reviewers. We appreciate the reviewers’ valuable comments and detailed responses to the comments are as follows:

Reviewer #1:

  1. The authors suggest that increased MMP1 and MMP13 activation by TG2 primarily leads to ECM degradation and not to fibrosis. However, TG2 can also affect/induce fibrosis in other pathways or by direct enzymatic action on ECM proteins and the relative effects of these consequences is still unkown. Have you observed any signs of fibrosis in dermal explant experiments? Has TG2 externalised in the dermal fibroblast culture? Please discuss in the discussion section.
  • Fibrosis is the result of chronic inflammatory response. Under our experimental conditions, accordingly, no sign of fibrosis was observed in dermal explant. Practically, it is very difficult to quantitate TG2 externalization. We are still thinking about the method to measure TG2 externalization. Previously, we showed that TG2-knockout mice exhibit reduced inflammation and fibrosis in response to bleomycin treatment. By contrast, the results in the present study showed that dsRNA-induced TG2 activation resulted in an increase of MMP-1 and -13 expression, which may lead to ECM degradation. These contradictory findings suggest that TG2-mediated crosslinking of collagen fibrils may limit MMP-1 or -13 activities, resulting in fragmented and coarsely distributed collagen fibrils as observed in skin chronically exposed to UV radiation. We added this point in the discussion. (Please, see lines 326-329 in the latest version of the manuscript for revision.)

  1. Reflecting also reference 13 where role of TG2 activation by UV was also shown in keratinocytes, which of the mechanisms (epidermal versus dermal in connective tissue fibroblasts) consider the authors more important for skin damage and how other transglutaminases also present in the skin epidermal layer can be involved as well, since in dermal explant experiments with TG2-/- cells still significant transglutaminase activity was present ?
  • In epidermal keratinocytes, TG1, TG2, TG3, TG5 and factor â…©IIIa are expressed. Studies using knockout mice for each TG isozyme revealed that TG1, TG3, and TG5 play a critical role in the formation of cornified envelope and cell-to-cell adhesion by crosslinking loricrin, envoplakin, periplakin, involucrin, small proline-rich proteins, and lipids, contributing to the skin barrier function including protection against UV radiation. These TG isozymes may be activated by an increase of intracellular calcium in response to UV-irradiation. In contrast, TG2 is expressed in both keratinocyte and dermal fibroblast, and elicits skin inflammation by inducing the secretion of inflammatory cytokines in both cell types through NF-kB activation. Thus, our results described in this study suggest that TG2-mediated MMP-1 and -13 expression may have a role in facilitating the migration of immune cells by fragmentation of collagen fibrils. To measure in situ (intracellular) TG activity in live cells, dermal explants are incubated in culture media containing biotinylated pentylamine (BP), an amine donor substrate incorporating into the cellular proteins, and then washed to remove unincorporated BP. After homogenization, cellular proteins crosslinked with BP are quantitated with HRP-streptavidin. Thus, the difference between wild-type and Tg2-knockout mice in the amount of BP-incorporated proteins is contributed by Tg2.

  1. Please describe more in details the specification (clone name) and binding characteristics of the non-commercial monoclonal anti-transglutaminase-2 antibodies used in the experiments. Which domain of TG2 they recognize? Was cross-reaction with other transglutaminases excluded? Does the antibody recognize both intracellular and extracellular TG2 and in which staining pattern was the TG2 protein expressed in the human dermal fibroblasts under basal conditions and after PolyI:C treatment ? Were any parallel experiments checked with other known commercial anti-TG2 antibodies like e.g. CUB7402 ?
  • Monoclonal antibody specific for TG2 (clone name: 3-3-1) was generated by immunizing Balb/c mice with the His-tagged TG2. After immunizing Balb/c mice, hybridoma cells were generated and selected [EMBO J, 2003. 22(19): p. 5273-82.]. This mAb for TG2 shows no cross-reactivity with recombinant TG1, TG3, TG4 and Factor XIIIa purified from Yeast as well as E. coli. Its Kd value is 10-8 ~ 10-9, but the antigen binding site has not been determined. Immunostaining of extracellular TG2 with 3-3-1 mAb has not been performed. I think it is difficult to discriminate intra- and extracellular TG2 using conventional immunostaining method. The specific binding of 3-3-1 mAb to TG2 was verified by immunoprecipitation and mass analysis, and further confirmed by comparison of TG2 immunostaining with commercially available anti-TG2 polyclonal antibody (Ab4, Thermo Fisher Scientific) with western blotting with 3-3-1 mAb (Please, see pages 21 in the latest version of the manuscript for revision.).

  1. Please specify in the methods that you used FITC-labeled DQ™-collagen type-I for the experiments. Both DQ™-collagen type-I and type-IV are commercially available.
  • We specify the DQ™-collagen “type-I” in the method section, as suggested (Please, see lines 99 and 394 in the latest version of the manuscript for revision.).

Reviewer 2 Report

In this manuscript, the authors present new pieces of evidence that poly(I:C) induces the transcription of MMP13 mRNA both in human and mouse dermal fibroblasts through signalling processes in which TG2 and ROS play a regulatory role. It is already known that MMPs are upregulated upon poly(I:C) treatment. But, this study tries to reveal more details about how TG2 and ROS play a regulatory role in the upregulation of MMP-13. In addition, the manuscript applies an ex vivo skin model. However, the discussion does not explain, digest and conclude the context of each experiment in a dynamic way which would be necessary to convince the readers about the existence of the hypothesised signalling process in the presented form.

One of the major concerns with the manuscript is that there is no Western blot demonstrating the changes of the MMP13 protein level in the case of in vitro experiments on HDF/MDF cells. Some Western blots should be incorporated in the figures.

The other major concern is that there is no experimental context, coherence between ROS level and TG2 activation in a time frame upon poly(I:C) treatment. The manuscript does not contain any explanation or speculation about the mechanism of TG2 activation by ROS. This should be added and discussed in the manuscript. In addition, if quercetin can eliminate the ROS production responsible for the upregulation of TG2 activity, the significantly lower ROS level and its time dependence should be demonstrated upon quercetin co-treatment in the MDF cells. Maybe the application of a NOX2 inhibitor could also be helpful.

The following comments should be addressed to improve the manuscript:

  1. The manuscript examined the effect of poly(I:C) only on MMP-13 (and at the beginning on MMP-1). The title is too general, and it should be more specific.
  2. Please, provide the number of independent repetitions of the experiments and the number of parallels which was used for the statistical analysis. The authors write in the method that statistical details are in the figure legends, but the exact type of the analysis in the case of given experiments frequently is missing. In the case of ANOVA, the applied ”post tests” should also be indicated.
  3. Figure 1c: Please, confirm your results using densitometry analysis in the case of MMP-1 and TG2 proteins level and demonstrate by adding a chart. The increase of MMP13 expression should be proved at the protein level here. The increased general MMPs activity on Fig1d does not verify the increase of MMP13 protein level.
  4. Fig 1e: Cystamine is not a specific TG2 inhibitor. The application of a specific one would be more convincing. This issue should also be addressed in the manuscript due to general readers not being familiar with the TG field.
  5. Lane 120 and 357. The description of in situ TG activity assay is controversial in the results and in the methods. Please correct it.
  6. Fig2d. Is it possible to demonstrate changes of TG2 protein level upon poly(I:C) treatment parallel with the TG2 activity? If TG2 protein level is constant, it would support the statement that TG2 is really activated.
  7. Fig3a and b. Demonstration of TG2 protein level in the experiments by Western blots would help to better conclude the experiment.
  8. Lane 160-163 and Fig4a: Please, show the ratio of ROS changes upon poly(I:C) treatment. If I am right, the ROS level is amplified more in the Tlr3-/- MDF than in WT, which could influence your hypothesis and sentences in lanes 160-163. Please, reconsider the sentence in lane 248.
  9. Lane 182-187: What is the mechanism of the quercetin effect? If it can decrease the level of ROS development, elimination of increased ROS level on Fig4a should be demonstrated to maintain the conclusion.

It is very unlikely that increased ROS level after 5 or 15 minutes (Fig4a-b) could be directly responsible for TG2 activation measured after 9 hr (Fig2d) or 12 hr Poly(I:C) treatment.

  1. What is the reason that 12 hr Poly(I:C) treatment causes various in situ TG2 activity level changes in the wild type MDF (see Fig 2d and Fig4e compared to Fig3a)
  2. Fig6d. Please, reconsider the visualisation of the TG2 regulatory role and particularly the effect of ROS during the revision.
  3. Fig6a. Please, complete the chart with densitometric analysis of other poly(I:C) treatments.
  4. Fig6b. What could be the explanation for the decreased TG2 level in the case of WT poly(I:C) treated samples (based on the Western blot TG2 protein level looks lower than on the untreated WT)? Densitometry analysis of TG2 levels could be helpful. Application of TG2 specific inhibitor could further convince the significance of TG2.
  5. Fig6. Have you tested the effect of specific ROS inhibition to prevent the increased expression of MMP-13 in the ex vivo skin model?
  6. Lane 320: Results with ex vivo skin culture were not presented in the case of Tlr3-/- mouse, please, remove from the sentence.
  7. Please, recheck the proper use of references. I tried to find the source of HDF and MDFs, and the text in lane 311 suggests that these were isolated by the authors. However, the HDF was probably purchased from GIBCO, if I am right? Ref41 is a review article about transglutaminases. Regarding anti-TG2 antibody, Ref43 cite another article as the source of the antibody. In the case of Western blot Ref13 (lane 346) contains another citation for sample preparation.
  8. Lane 374-379. Please, provide more details about the ROS measurement, e.g. preparation of DCF-DA solution, way of cell harvest.
  9. Please, give the vector source for the luciferase assays more accurately. In the case of vectors from Addgene, acknowledge the vector using the necessary text into the methods and the reference from the Addgene webpage of the vector.

Minor comments:

On some charts, the SD or SED error bars are not visible.

Based on the repetition number, please complete the „original images” file. If it is possible, show the whole membrane in this image collection.

Mouse TG2 gene is labelled by Tgm2, please, reconsider the usage of this abbreviation.

Author Response

Feb 20, 2020

Simone Beninati

Guest Editor

Dear Dr. Simone Beninati:

We are pleased to submit to you a revised manuscript entitled “Double-stranded RNA enhances matrix metalloproteinase-1 and -13 expressions through TLR3-dependent activation of transglutaminase 2 in dermal fibroblasts” [Manuscript ID: ijms-1588932]. The manuscript has been changed along the line suggested by reviewers. We appreciate the reviewers’ valuable comments and detailed responses to the comments are as follows:

Reviewer #2:

  1. One of the major concerns with the manuscript is that there is no Western blot demonstrating the changes of the MMP13 protein level in the case of in vitro experiments on HDF/MDF cells. Some Western blots should be incorporated in the figures.
  • Western blot demonstrating the changes of the Mmp13 protein level has been added in Figure 2b, as suggested. Under same experimental conditions, Tgm2-/- MDFs showed significantly reduced level of Mmp13 protein compared with wild-type MDFs.

  1. The other major concern is that there is no experimental context, coherence between ROS level and TG2 activation in a time frame upon poly(I:C) treatment. The manuscript does not contain any explanation or speculation about the mechanism of TG2 activation by ROS. This should be added and discussed in the manuscript. In addition, if quercetin can eliminate the ROS production responsible for the upregulation of TG2 activity, the significantly lower ROS level and its time dependence should be demonstrated upon quercetin co-treatment in the MDF cells. Maybe the application of a NOX2 inhibitor could also be helpful.
  • Our previous studies showed that TG2 is activated under various oxidative stress conditions, including UV irradiation, ROS-generating chemotherapeutics and H2O2 treatment [1-6]. Moreover, poly(I:C) treatment generates ROS through TLR3 binding. In this study, therefore, we tested whether poly(I:C) treatment could activate TG2. Oxidative stress activates TG2 by one of two mechanisms; (i) by releasing endoplasmic reticulum (ER) calcium [6], and (ii) by activating TGFb signaling pathway [1] or possibly redox signaling pathway, depending on the level of oxidative stress. Moreover, TG2 is not activated when treated with actinomycin D after oxidative stress. These results suggest that TG2 is activated by TGFb or redox signaling-mediated regulation of gene expression, explaining the time interval between ROS production and TG2 activation. Thus, in the poly(I:C) co-treatment experiment, time dependent effect of quercetin, a flavonoid widely used as an antioxidant, on in situ TG2 activity and MMP13 expression can not be evaluated. These points have been described in the introduction. (Please, see lines 269-273 / lines 70-75 in the latest version of the manuscript for revision.)

  1. The manuscript examined the effect of poly(I:C) only on MMP-13 (and at the beginning on MMP-1). The title is too general, and it should be more specific.
  • In the title, “matrix metalloproteinase expression” has been changed to “matrix metalloproteinase-1 and -13 expressions” as suggested.

  1. Please, provide the number of independent repetitions of the experiments and the number of parallels which was used for the statistical analysis. The authors write in the method that statistical details are in the figure legends, but the exact type of the analysis in the case of given experiments frequently is missing. In the case of ANOVA, the applied ”post tests” should also be indicated.
  • The number of independent repetitions of each experiment and the applied post-test of each statistical analysis are added in figure legends.

  1. Figure 1c: Please, confirm your results using densitometry analysis in the case of MMP-1 and TG2 proteins level and demonstrate by adding a chart. The increase of MMP13 expression should be proved at the protein level here. The increased general MMPs activity on Fig1d does not verify the increase of MMP13 protein level.
  • In Figure 1c, the ratio of MMP-1/Actin or TG2/Actin in each blot calculated by densitometry analysis is added. Western blot of Mmp13 and the ratio of Mmp13/Actin are added in Figure 2c.

  1. Fig 1e: Cystamine is not a specific TG2 inhibitor. The application of a specific one would be more convincing. This issue should also be addressed in the manuscript due to general readers not being familiar with the TG field.
  • The role of TG2 in the regulation of dsRNA-induced MMP-1 and MMP-13 expression has been demonstrated by the experiments using TG2 siRNA and knock-out mouse. Although cystamine is a nonspecific TG2 inhibitor, the effect of cystamine has been tested to show further pharmacologic application of TG2 inhibitor. Moreover, TG2 is the only isoform expressed in dermal fibroblasts, and TG2 specific inhibitor is not available. Characteristics of cystamine for general reader is described in the results. (Please, see lines 103-104 in the latest version of the manuscript for revision.)

  1. Lane 120 and 357. The description of in situ TG activity assay is controversial in the results and in the methods. Please correct it.
  • The description of assay procedure in the results has been changed as indicated. (Please, see lines 129-130 in the latest version of the manuscript for revision.)

  1. Is it possible to demonstrate changes of TG2 protein level upon poly(I:C) treatment parallel with the TG2 activity? If TG2 protein level is constant, it would support the statement that TG2 is really activated.
  • Western blot for Tg2 protein after poly(I:C) treatment is added in Fig.2E. Tg2 protein level is gradually increased by poly(I:C) treatment, but is not well-correlated with in situ TG activity (Please, see lines 133-135 in the latest version of the manuscript for revision.). Indeed, Tg2 is held in a latent inactive state but is activated in response to various stresses (see also response of comment #2). In situ TG2 activity is regulated by competitive binding of calcium and magnesium ion to calcium binding sites [7]. Moreover, unidentified inhibitor or activator regulated by TGFb and redox signaling may be involved in the TG2 activation, and could explain the discrepancy between TG protein level and in situ activity.

  1. Fig3a and b. Demonstration of TG2 protein level in the experiments by Western blots would help to better conclude the experiment.
  • Western blot of Tg2 protein is added in Fig. 3C. In WT and Tlr3-/- MDFs, poly(I:C) had no statistically significant effect on Tg2 protein and mRNA level compared with saline-treated cells. The result has been changed accordingly (Please, see lines 158-160 in the latest version of the manuscript for revision.).

  1. Lane 160-163 and Fig4a: Please, show the ratio of ROS changes upon poly(I:C) treatment. If I am right, the ROS level is amplified more in the Tlr3-/- MDF than in WT, which could influence your hypothesis and sentences in lanes 160-163. Please, reconsider the sentence in lane 248.
  • Figure 4a is replotted with relative ratio of ROS level as suggested. Control Tlr3-/- MDFs exhibit significant low ROS levels compared with control wild-type MDFs. However, poly(I:C) treatment had no effect on ROS level of Tlr3-/- MDFs, but resulted in a gradual increase of ROS level of wild-type MDFs. These results suggest that poly(I:C) treatment generates ROS by binding to TLR3 and by unknown mechanism. Therefore, the sentence is modified (Please, see lines 176-180 in the latest version of the manuscript for revision.).

  1. Lane 182-187: What is the mechanism of the quercetin effect? If it can decrease the level of ROS development, elimination of increased ROS level on Fig4a should be demonstrated to maintain the conclusion. It is very unlikely that increased ROS level after 5 or 15 minutes (Fig4a-b) could be directly responsible for TG2 activation measured after 9 hr (Fig2d) or 12 hr Poly(I:C) treatment.
  • Quercetin is a polyphenolic flavonoid and exhibits antioxidant effects. Although roles in the regulation of ROS level, signaling pathways, glutathione synthesis, and redox-sensitive enzymes have been suggested, exact mechanism for antioxidant effects of quercetin is unclear. Our previous report showed that in situ TG2 activity of lens epithelial cells is increased and peaked around 12 h after UV-irradiation or H2O2 treatment [1] through activation of TGFb signaling pathway [8]. In dermal fibroblasts as shown in this study, in situ TG2 activity is peaked at 9 h - 12 h after poly(I:C) treatment. We are currently scrutinizing redox signaling as well as TGFb signaling.

  1. What is the reason that 12 hr Poly(I:C) treatment causes various in situ TG2 activity level changes in the wild type MDF (see Fig 2d and Fig4e compared to Fig3a)
  • To measure in situ (intracellular) TG activity in live cells, cells are incubated in culture media containing biotinylated pentylamine (BP), an amine donor substrate incorporating into the cellular proteins, and then washed to remove unincorporated BP. After homogenization, cellular proteins crosslinked with BP are quantitated with HRP-streptavidin. Thus, in situ TG activity is affected by many factors, such as cell confluency. This performance characteristics of the assay is main reason for various in situ TG activity levels. Thus, relative level of in situ activity could be compared within each experiment.

  1. Please, reconsider the visualisation of the TG2 regulatory role and particularly the effect of ROS during the revision.
  • Reflecting your comment, we revised our schematic representation (Please, see fig 6d).

  1. Please, complete the chart with densitometric analysis of other poly(I:C) treatments.
  • The chart with densitometric analysis of western blotting is completed.

  1. What could be the explanation for the decreased TG2 level in the case of WT poly(I:C) treated samples (based on the Western blot TG2 protein level looks lower than on the untreated WT)? Densitometry analysis of TG2 levels could be helpful. Application of TG2 specific inhibitor could further convince the significance of TG2.
  • Levels of TG2 protein in poly(I:C)-treated wild-type mice appeared to be lower than those of control mice. Densitometry analyses were performed (added in Fig. 6b) as suggested, and statistical significance was tested by two-way ANOVA by Turkey’s post-test. There was no statistically significant difference in TG2/actin ratio between control and poly(I:C)-treated mice. TG2-specific inhibitor for assessing in situ activity is not currently available.

  1. Have you tested the effect of specific ROS inhibition to prevent the increased expression of MMP-13 in the ex vivo skin model?
  • No, we haven’t tested. Although many newer antioxidant agents are now under development, effective, specific ROS-reducing agent is not available.

  1. Lane 320: Results with ex vivo skin culture were not presented in the case of Tlr3-/- mouse, please, remove from the sentence.
  • The sentence about Tlr3 -/- mouse is deleted (Please, see lines 348-349 in the latest version of the manuscript for revision.).

  1. Please, recheck the proper use of references. I tried to find the source of HDF and MDFs, and the text in lane 311 suggests that these were isolated by the authors. However, the HDF was probably purchased from GIBCO, if I am right? Ref41 is a review article about transglutaminases. Regarding anti-TG2 antibody, Ref43 cite another article as the source of the antibody. In the case of Western blot Ref13 (lane 346) contains another citation for sample preparation.
  • Your comment is correct. HDFs were purchased from GIBCO. MDFs were isolated from neonatal mice skin as previously described (Ref14, lines 338-340). Ref13 for Western blot analysis and Ref for anti-TG2 antibody are changed to new references (Please see lines 372-373 / lines 378-379).

  1. Lane 374-379. Please, provide more details about the ROS measurement, e.g. preparation of DCF-DA solution, way of cell harvest.
  • Detailed information about intracellular ROS measurement is added in the ‘method’ (Please, see lines 401-411 in the latest version of the manuscript for revision.).

  1. Please, give the vector source for the luciferase assays more accurately. In the case of vectors from Addgene, acknowledge the vector using the necessary text into the methods and the reference from the Addgene webpage of the vector.
  • Detailed information about vector source and reference from Addgene are added as you suggested (Please, see lines 414-416 in the latest version of the manuscript for revision.).

  1. On some charts, the SD or SED error bars are not visible.
  • Statistical analysis is repeatedly conducted and error bars in every figure are checked.

  1. Based on the repetition number, please complete the „original images” file. If it is possible, show the whole membrane in this image collection.
  • Original and unadjusted whole membrane images are uploaded (Please, see pages 17-20 in the latest version of the manuscript for revision.).

  1. Mouse TG2 gene is labelled by Tgm2, please, reconsider the usage of this abbreviation.
  • Tg2 has been changed to Tgm2.

References

  1. Shin, D.M.; Jeon, J.H.; Kim, C.W.; Cho, S.Y.; Kwon, J.C.; Lee, H.J.; Choi, K.H.; Park, S.C.; Kim, I.G. Cell type-specific activation of intracellular transglutaminase 2 by oxidative stress or ultraviolet irradiation: implications of transglutaminase 2 in age-related cataractogenesis. J Biol Chem 2004, 279, 15032-15039, doi:10.1074/jbc.M308734200.
  2. Cho, S.Y.; Jeong, E.M.; Lee, J.H.; Kim, H.J.; Lim, J.; Kim, C.W.; Shin, D.M.; Jeon, J.H.; Choi, K.; Kim, I.G. Doxorubicin induces the persistent activation of intracellular transglutaminase 2 that protects from cell death. Mol Cells 2012, 33, 235-241, doi:10.1007/s10059-012-2201-9.
  3. Jeong, E.M.; Kim, C.W.; Cho, S.Y.; Jang, G.Y.; Shin, D.M.; Jeon, J.H.; Kim, I.G. Degradation of transglutaminase 2 by calcium-mediated ubiquitination responding to high oxidative stress. FEBS Lett 2009, 583, 648-654, doi:10.1016/j.febslet.2009.01.032.
  4. Shin, J.W.; Kwon, M.A.; Hwang, J.; Lee, S.J.; Lee, J.H.; Kim, H.J.; Lee, K.B.; Lee, S.J.; Jeong, E.M.; Chung, J.H.; et al. Keratinocyte transglutaminase 2 promotes CCR6(+) gammadeltaT-cell recruitment by upregulating CCL20 in psoriatic inflammation. Cell Death Dis 2020, 11, 301, doi:10.1038/s41419-020-2495-z.
  5. Oh, K.; Park, H.B.; Byoun, O.J.; Shin, D.M.; Jeong, E.M.; Kim, Y.W.; Kim, Y.S.; Melino, G.; Kim, I.G.; Lee, D.S. Epithelial transglutaminase 2 is needed for T cell interleukin-17 production and subsequent pulmonary inflammation and fibrosis in bleomycin-treated mice. J Exp Med 2011, 208, 1707-1719, doi:10.1084/jem.20101457.
  6. Lee, S.J.; Lee, K.B.; Son, Y.H.; Shin, J.; Lee, J.H.; Kim, H.J.; Hong, A.Y.; Bae, H.W.; Kwon, M.A.; Lee, W.J.; et al. Transglutaminase 2 mediates UV-induced skin inflammation by enhancing inflammatory cytokine production. Cell Death Dis 2017, 8, e3148, doi:10.1038/cddis.2017.550.
  7. Jeong, E.M.; Lee, K.B.; Kim, G.E.; Kim, C.M.; Lee, J.H.; Kim, H.J.; Shin, J.W.; Kwon, M.A.; Park, H.H.; Kim, I.G. Competitive Binding of Magnesium to Calcium Binding Sites Reciprocally Regulates Transamidase and GTP Hydrolysis Activity of Transglutaminase 2. Int J Mol Sci 2020, 21, doi:10.3390/ijms21030791.
  8. Shin, D.M.; Jeon, J.H.; Kim, C.W.; Cho, S.Y.; Lee, H.J.; Jang, G.Y.; Jeong, E.M.; Lee, D.S.; Kang, J.H.; Melino, G.; et al. TGFbeta mediates activation of transglutaminase 2 in response to oxidative stress that leads to protein aggregation. FASEB J 2008, 22, 2498-2507, doi:10.1096/fj.07-095455.

Round 2

Reviewer 2 Report

The manuscript has been significantly improved. However, the explanation of the poly(I:C) induced Tlr3 dependent ROS generation based on the presented results were still not completely correct.

In Fig4A., the authors demonstrate that in Tlr3 KO MDF, Poly(I:C) treatment increases approximately 2 or 2.5 times the ROS level. But, in the text, they write: „poly(I:C) treatment had no effect on ROS level of Tlr3-/- MDFs, but resulted in a gradual increase of ROS level of wild-type MDFs (Figure 4A)” (lane 178-180). This statement looks controversial with the obvious change on the chart in the case of Tlr3-/- MDF (Fig. 4A), and it would be better to modify the text based on the measured evidence.

In addition, in the light of this experiment, the next statement in the discussion: „Moreover, poly(I:C) treatment generates ROS through TLR3 binding. In this study, therefore, we tested whether poly(I:C) treatment could activate TG2. Our results showed that binding of poly(I:C) to TLR3 generates ROS, which activates TG2 in dermal fibroblast” (lane 272-275) was not fully supported/confirmed by the results presented in this study. The demonstration of poly(I:C) induced Tlr3 dependent ROS generation is not convincing in this study. The sentences should be modified and/or completed with references. Based on the results, it is obvious, that ROS contributes to the increased Mmp13 expression in a Tg2 dependent way, but the experiment which would prove poly(I:C) induced, Tlr3 dependent ROS generation in MDF is still missing from this work.

The correlated sentence in the abstract could also be modified.

Please, check the spelling in the figures (Fig1d, 2c).

Author Response

Feb 25, 2020

Simone Beninati

Guest Editor

Dear Dr. Simone Beninati:

We are pleased to submit to you a revised manuscript entitled “Double-stranded RNA enhances matrix metalloproteinase-1 and -13 expressions through TLR3-dependent activation of transglutaminase 2 in dermal fibroblasts” [Manuscript ID: ijms-1588932]. The manuscript has been changed along the line suggested by reviewers. We appreciate the reviewers’ valuable comments and detailed responses to the comments are as follows:

Reviewer #2:

  1. In Fig4A., the authors demonstrate that in Tlr3 KO MDF, Poly(I:C) treatment increases approximately 2 or 2.5 times the ROS level. But, in the text, they write: „poly(I:C) treatment had no effect on ROS level of Tlr3-/-MDFs, but resulted in a gradual increase of ROS level of wild-type MDFs (Figure 4A)” (lane 178-180). This statement looks controversial with the obvious change on the chart in the case of Tlr3-/- MDF (Fig. 4A), and it would be better to modify the text based on the measured evidence.
  • We have changed the text like following description; “Poly(I:C) treatment had increased the ROS level gradually in both MDFs, but the ROS level in Tlr3-/-MDFs had still held lower than that of wild-type MDFs (Figure 4A). These results suggest that poly(I:C) treatment generates ROS by binding to TLR3 and by unknown mechanism.” (Please, see highlighted lines 177-180 in the latest version of the manuscript for revision.)

  1. In addition, in the light of this experiment, the next statement in the discussion: „Moreover, poly(I:C) treatment generates ROS through TLR3 binding. In this study, therefore, we tested whether poly(I:C) treatment could activate TG2. Our results showed that binding of poly(I:C) to TLR3 generates ROS, which activates TG2 in dermal fibroblast” (lane 272-275) was not fully supported/confirmed by the results presented in this study. The demonstration of poly(I:C) induced Tlr3 dependent ROS generation is not convincing in this study. The sentences should be modified and/or completed with references. Based on the results, it is obvious, that ROS contributes to the increased Mmp13 expression in a Tg2 dependent way, but the experiment which would prove poly(I:C) induced, Tlr3 dependent ROS generation in MDF is still missing from this work.
  • We have changed the text like following description; “Moreover, poly(I:C) treatment generates ROS through TLR3 binding in macrophages [17]. In this study, therefore, we tested whether poly(I:C) treatment could activate TG2. Our results showed that poly(I:C) generates ROS by binding to TLR3 and by unknown mechanism, which activates TG2 in dermal fibroblast.” (Please, see highlighted lines 271-274 in the latest version of the manuscript for revision.)

  1. The correlated sentence in the abstract could also be modified.
  • We have edited the correlated sentence in the abstract. (Please, see highlighted lines 23-24 in the latest version of the manuscript for revision.)

  1. Please, check the spelling in the figures (Fig1d, 2c).
  • The title of Y axis in Figure 1D and 2C has been changed correctly.

I hope our responses could convince reviewers.

Sincerely yours,

In-Gyu Kim, M.D., Ph.D.

Department of Biochemistry and Molecular Biology

Seoul National University College of Medicine
